# Explainable AI for sign language recognition models: Integrating Grad-Cam LIME and Integrated Gradients

**Fatima-Zahrae El-Qoraychy**[1]*, **Yazan Mualla**[1], **Hui Zhao**[2], **Mahjoub Dridi**[1], **Jean-Charles Créput**[1], **Luca Longo**[3,4]

**1** Université de Technologie de Belfort Montbéliard, UTBM, CIAD UR 7533, Belfort, France, **2** Department of Computer Science and Technology, Tongji University, Shanghai, China, **3** Artificial Intelligence and Cognitive Load Research Lab, University College Cork, Cork, Ireland, **4** School of Computer Science and Information Technology, University College Cork, Western Gateway Building, Cork, Ireland

* fatima.el-qoraychy@utbm.fr

## Abstract

Sign language recognition is crucial in bridging the communication gaps between hearing and deaf communities. In this study, we build on an existing sign language classification model based on the VGG19 architecture, enhancing its robustness through dataset augmentation and alternative data representations. We introduce a segmentation-based approach that utilizes hand masks generated by the U-Net model, replacing depth images to mitigate noise and improve classification accuracy. We use an Explainable Artificial Intelligence (XAI) approach that incorporates Grad-CAM, LIME, and integrated gradients methods to interpret the model's decision-making process, ensuring transparency and reliability in real-world applications. Our comparative analysis between Red-Green-Blue (RGB) and mask-based models demonstrates that while the RGB model benefits from richer texture and color information, the mask-based model effectively focuses on hand shape and structure. The integration of XAI further validates our results by highlighting key regions of the image that influence the model's predictions and by enabling a multi-perspective analysis that captures complementary aspects of the input, including region-based attention, pixel-level attribution, and structural shape analysis, thereby facilitating a deeper understanding of the model's internal representations and potential failure modes. This work contributes to advancing American Sign Language recognition by improving model generalisation and explainability, ultimately fostering greater trust and usability in assistive technologies.

## Introduction

Human-Computer Interaction (HCI) is based on exchanging information and commands between human users and technological systems. It explores how humans

**Data availability statement:** Our code and dataset are available at the following links: https://www.kaggle.com/code/fatimaelqoraychy/sign-language-classification-cnn-vgg19-1 https://www.kaggle.com/datasets/fatimaelqoraychy/dataset. This information has been included in the manuscript.

**Funding:** Publication fees were supported by the Artificial Intelligence and Cognitive Load Research Lab and the School of Computer Science and Information Technology, University College Cork, Ireland [Funding to L.L.]. The funders had no role in study design, data collection, decision to publish, or manuscript writing.

**Competing interests:** The authors have declared that no competing interests exist.

interact with various technologies, including computers, machines, artificial intelligence (AI), agents, and robots. These interactions can take multiple forms: cooperation, collaboration, team dynamics, symbiosis, and integration. Understanding these relationships is essential for designing systems that enhance user experience while improving the transparency and trustworthiness of technological systems [1]. The early studies on HCI primarily focused on designing user-friendly interfaces for command-line systems [2]. Over time, HCI has expanded to include multimodal interaction methods such as touch interfaces, voice recognition, and gesture-based controls [3]. The growing presence of computing devices, from personal computers to smartphones, makes HCI ubiquitous in everyday life. This increased reliance on various interaction forms requires continuous innovation to ensure technology remains intuitive, accessible, and efficient for users. There are various forms of HCI, such as text input, voice commands, and gestures. Text-based inputs, like typing on a keyboard, remain fundamental but inefficient, especially for users with physical or cognitive impairments. Voice recognition technologies, such as virtual assistants (Siri, Alexa, etc.), offer hands-free interaction, valuable in contexts where manual input is impractical [4]. Gesture recognition, which underpins Sign Language Recognition (SLR), is vital in bridging the communication gap between the deaf or hard-of-hearing community and the hearing population by converting gestures into text or speech. Early approaches to SLR used rule-based systems or sensor-based inputs. However, advances in AI, particularly Convolutional Neural Networks (CNNs), have enabled the development of more sophisticated models for recognising sign language from video sequences or images [5–7]. These models significantly improve accessibility for individuals with hearing impairments, promoting inclusivity across various sectors, including education and customer service. Despite these advancements, deploying AI-driven SLR systems remains challenging due to deep learning models' "black box" nature [8]. While CNNs excel in classification tasks, the inferential mechanism they learn from input to output remains opaque. This makes it difficult for developers and users to understand why a model behaves in certain ways, especially when misclassifications occur. This limited transparency raises concerns about trust, reliability, and adaptability, particularly when dealing with complex gestures, individual variations, or regional differences in sign language. For example, signs may vary due to factors like signing speed, personal style, or regional dialects, further complicating classification. Thus, addressing these challenges requires enhancing the AI models' explainability.

Explainable Artificial Intelligence (XAI) is key to overcoming this barrier. XAI includes techniques designed to make AI systems more interpretable by clearly explaining their decisions and actions [9,10]. In the context of SLR, XAI can help identify which gesture features influence a model's prediction, allowing users and developers to better understand and improve the system. Moreover, XAI ensures that AI systems are fair, transparent, and free from hidden biases, which is particularly important in SLR, where accessibility and inclusivity are paramount. It is increasingly recognised that fostering effective communication and understanding between humans and AI systems is vital for their successful integration into various applications [11,12]. Explainability is essential for technical improvement and fostering trust with users who rely on these systems for critical communication.

The Defence Advanced Research Projects Agency (DARPA) launched the "XAI Program" in 2017 [13], sparking significant advancements in AI explainability. Notable contributions include the HAExA architecture [14], which provides transparent agent decision explanations, while RISE [15] generates importance maps to highlight key neural network elements. These initiatives underscore the growing importance of XAI across various domains, including SLR, where it improves model performance and trustworthiness by offering insights into decision-making processes. Several techniques have emerged in XAI, each offering distinct advantages depending on the application. Methods such as Local Interpretable Model-Agnostic Explanations (LIME) [16–19], SHapley Additive exPlanations (SHAP) [20–22], and Gradient-weighted Class Activation Mapping (Grad-CAM) [23–26] have been widely used to explain AI inferential mechanisms in various domains.

In this article, we explore how these techniques can be applied to interpret CNN-based SLR systems, focusing on enhancing both model transparency and performance. The primary objective of this work is to develop a real-time prediction system for American Sign Language (ASL) alphabet recognition, as illustrated in Fig 1 (the referenced ASL alphabet chart is extracted from [27]). The system is based on RGB and mask images to predict letters, while integrating XAI techniques to improve the interpretation of the input–output mapping learned by the models. Our proposed approach combines three complementary explainability techniques, each offering distinct insights, as illustrated in Fig 2. These include:

- Region-based Explanation: it highlights key areas in an input image that strongly influence classification outcomes;
- Pixel-Level Importance: it evaluates individual pixel contributions to the final decision;
- Shape Analysis: it focuses on structural elements, understanding how variations in form impact classification results.

The structure of this paper is as follows. Sect 2 presents an overview of the basic concept of SLR, existing classification models, and relevant XAI techniques. Sect 3 outlines the experimental setup, detailing how the classification and XAI models are integrated. In Sect 4, a discussion of the results, focusing on the insights gained through XAI and the improvements made to the classification model, is presented. Eventually, Sect 5 concludes the article by summarizing the contributions and proposing future research directions.

## Related work

SLR has attracted growing interest due to its potential to bridge communication barriers for the deaf and hard-of-hearing communities. Recent advances in deep learning, particularly CNNs, have significantly improved the performance of SLR systems, enabling the automatic recognition of hand gestures from RGB images, videos, and depth data [28–31].

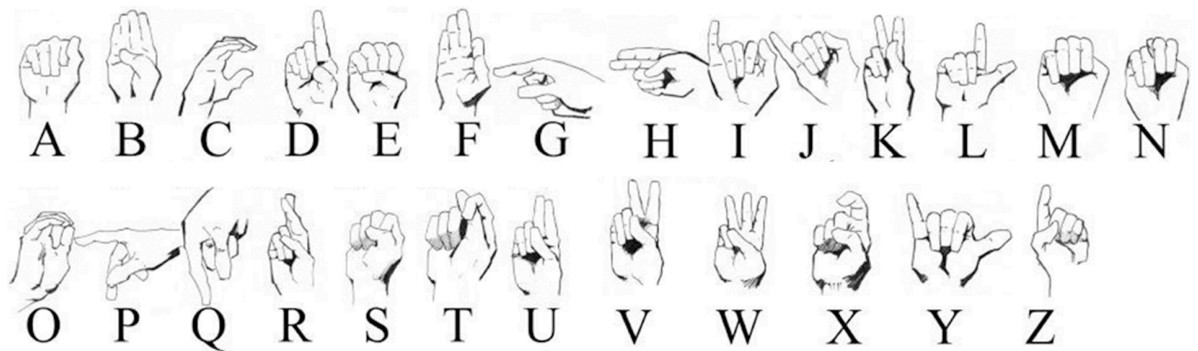

**Fig 1**. American sign language alphabet (based on [27]).

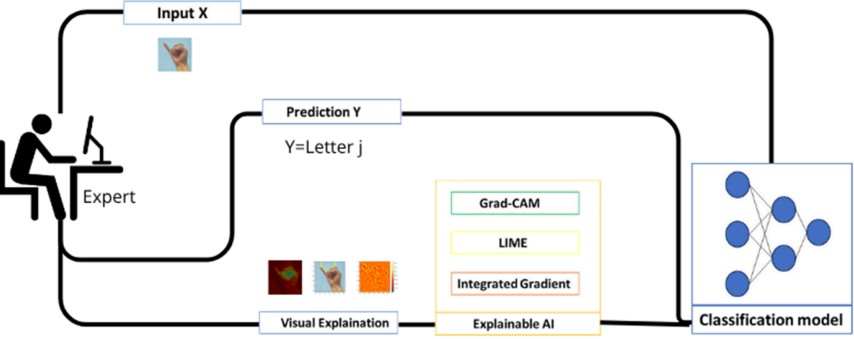

**Fig 2**. Real-time ASL recognition approach with integrated visual explanation using Grad-CAM, LIME, and Integrated Gradients.

Early works focused primarily on handcrafted features and skin colour segmentation in various colour spaces such as RGB and HSV [32]. These methods improved gesture localisation but were highly sensitive to lighting variations, background clutter, and inter-user variability. For example, Tripathi et al. [33] developed a gesture-to-text translation system for Indian Sign Language using speeded-up robust features for feature extraction, combined with edge detection, skin masking, and a Bag of Visual Words model followed by classical machine learning classifiers. Similarly, [34] proposed an SVM-based recognition system trained on a custom dataset. While effective in structured settings, these traditional approaches do not scale well to complex or dynamic gestures. In parallel, some early approaches leveraged sensor-based systems to capture hand kinematics directly, bypassing the need for visual input. For example, Wang et al. [35] proposed an ASL recognition system that combined data from a CyberGlove and a Flock of Birds motion tracker within a multi-dimensional Hidden Markov Model (HMM) framework, enabling the recognition of both static and dynamic gestures. Similarly, Raheem [27] utilised a sensory glove in conjunction with a Multi-Layer Perceptron (MLP) to classify ASL hand gestures and analyse recognition accuracy. While both systems demonstrated high performance in controlled conditions, they share key limitations: a reliance on costly and intrusive hardware, poor generalisation to diverse signers or environments, and limited scalability beyond laboratory settings. Furthermore, these sensor-dependent approaches are tightly coupled to specific modalities, making them less adaptable to current vision-based, marker-free SLR trends. To overcome such limitations, researchers have explored skeletal data obtained from depth sensors like Microsoft Kinect [36,37], which capture 3D joint coordinates for robust spatio-temporal modelling. Although these methods enhance recognition of dynamic signs, they require specialised hardware, limiting their use in real-world or low-resource environments. More recently, deep learning approaches have emerged as the dominant paradigm. For example, Talaat et al. [38] proposed a real-time Arabic Sign Language translation system using YOLOv8 for gesture detection and an animated avatar for visual feedback, achieving high accuracy across multiple datasets. Najib [39] developed MSLI, a multilingual system capable of recognising signs from 11 different sign languages using a two-stage pipeline for language detection and gesture classification, demonstrating strong generalisation capabilities. In the context of real-time applications, CNN-based pipelines have been successfully deployed [40,41]. However, many of these models rely on large annotated datasets that often lack diversity in terms of hand shape, skin tone, and signing styles [42,43]. As a result, generalisation to heterogeneous populations remains a challenge. Moreover, deep learning models often act as black boxes, limiting transparency and raising concerns about fairness and reliability, especially in assistive or safety-critical settings. In sign language, subtle gesture variations can alter meaning, amplifying the need for interpretable models. Recent studies have integrated Explainable Artificial Intelligence (XAI) techniques into SLR frameworks [44–46]. Methods such as LIME, SHAP, and Grad-CAM help visualise the relationship between input features and model predictions. McCleary et al. [47] utilised micro-Doppler radar signatures and implemented a custom explainability algorithm to localise relevant signal regions. While their method

achieves impressive accuracy using a compact dataset, the explainability remains limited to signal-domain saliency. Ridwan et al. [48] employed transfer learning and SHAP-based interpretability in a multi-model architecture, demonstrating enhanced trust in gesture classification systems. While existing works provide valuable contributions, most rely on a single post-hoc XAI method and do not offer comparative evaluations or explore the effect of explanation strategies across varying gesture complexities. Furthermore, although some studies use XAI to uncover biases or improve debugging, there remains a lack of methodological frameworks that integrate explainability into the model design process itself rather than treating it as a post-hoc diagnostic tool. This gap hinders the development of more trustworthy and generalizable SLR systems as presented in the review [49,50]. To provide a comprehensive overview of prior work, Table 1 presents a chronological summary of representative SLR systems, detailing their methodologies, the incorporation of explainability techniques, and the reported performance. This selection covers a wide range of approaches, from traditional handcrafted pipelines and rule-based segmentation to CNNs, skeleton-based models, and classical machine learning techniques. It also reflects the diversity of input modalities (e.g., RGB, depth, glove-based sensors, and segmented images) and the application of XAI methods, such as modular feedback, LIME, and SHAP, either as post-hoc analysis tools or integrated components within the training and validation process.

Despite notable progress in SLR, many existing systems still face key limitations, including limited generalisation to diverse users and environments and a lack of interpretability in model predictions. Moreover, most prior works employ a single explainability technique as a post-hoc diagnostic tool, without exploring how different methods might complement each other. To address these gaps, our work proposes a real-time prediction system for ASL alphabet recognition, leveraging both RGB and segmentation-based representations. In contrast to traditional colour-based segmentation methods, which are often sensitive to lighting variations and skin tone, we employ a U-Net-based segmentation model to generate robust binary hand masks. This enables more consistent gesture localisation across varied backgrounds and user profiles, enhancing generalisation. Furthermore, we integrate three distinct XAI methods (Grad-CAM, LIME, and integrated gradients) to provide interpretability and to analyse their respective strengths in capturing region-level, pixel-level, and structural information. By applying these techniques systematically to the same inputs, we demonstrate how their complementary perspectives can offer a richer understanding of model behaviour, highlight failure cases, and guide model refinement.

**Table 1**. Chronological Summary of SLR Methods, Explainability, and Reported Accuracy.

| Ref. | Year | Method Used | Explainability | Accuracy |
|------|------|-------------|----------------|----------|
| [32] | 2014 | segmentation + rule-based | No | 90% |
| [41] | 2018 | CNN | No | 91% |
| [36] | 2018 | CNN + skeletal data | No | 91.28% |
| [27] | 2019 | Sensory glove + recognition networks | No | 90.19% |
| [40] | 2019 | SLRNet-8 | No | 99.90% |
| [46] | 2019 | CNN | modular feedback | 87.9% |
| [29] | 2020 | CNN + Transformer | No | 93.33% |
| [42] | 2021 | CNN | No | 98.96% |
| [47] | 2021 | CNN + micro-Doppler | modular feedback | 90.3% |
| [33] | 2023 | SURF + BoVW + SVM | No | 92% |
| [45] | 2023 | ResNet50 + self-attention | LIME + SHAP | 98.2% |
| [38] | 2023 | YOLOv8 + Animated Avatar | No | 99.4% |
| [34] | 2024 | SVM + MediaPipe | No | 99.8% |
| [48] | 2024 | ResNet | SHAP | 98.9% |
| [50] | 2024 | XGBoost + ExtraTrees + Random Forest | LIME + SHAP | 92.99% |
| [39] | 2024 | MSLI system | No | 92.3% |

## Methodology

The starting point for the development of an explainable SLR system is the work developed by Damion Joyner (https://www.kaggle.com/code/damionjoyner/sign-language-classification-cnn-vgg19), which endeavours to classify a set of RGB and depth images of ASL using a CNN model based on the Visual Geometry Group 19 (VGG19) architecture. This model is trained using the ASL alphabet dataset, which comprises over 100,000 images of English letters in sign language collected from five individuals. With 24 letters in the English alphabet and the images provided by five pairs of hands, the model must be capable of classifying images based on the different letters.

## Classification model structure

The classification model is based on a combination of the pre-trained VGG19 model, enriched with additional layers designed specifically for the image classification task. The CNN architecture devised by the Visual Geometry Group (VGG) is widely recognised for its depth, characterised by many convolution layers. It has played a key role in the evolution of object recognition models, outperforming many benchmarks on various tasks and datasets. It is pretrained with ImageNet datasets and can still outperform unseen datasets, making it one of the most used image recognition architectures. Multiple variants exist for such a network, including VGG-16 and VGG-19, differing only in the total number of layers. VGG19 is an advanced version of the VGG architecture, incorporating 19 convolution layers [51]. It consists of several convolutional blocks, each comprising multiple convolutional layers followed by pooling layers. The model utilises small filters (3×3) with a stride pattern of 1 and padding of 1 to preserve extracted feature sizes. Using this pattern for the convolutional layers means that the convolutional filters move one pixel at a time across the input data, and one layer of zero pixels is added around the input to maintain its size during the convolution process. After the convolutional blocks, the network connects to fully connected layers for classification. While the VGG19 architecture is relatively established and widely used in image classification tasks, it was chosen for this project due to its depth, simplicity, and ability to generalise across unseen datasets. VGG19 remains a competitive architecture for tasks involving image recognition, and its extensive use allows for leveraging pretrained weights on ImageNet, speeding up training and improving accuracy. In this work, the classification model comprises both the VGG19 model and additional layers. The VGG19 layers adhere to the standard architecture, featuring blocks of convolutional layers followed by pooling layers. These layers progressively capture features across different scales and complexities. Additional layers are then introduced after the VGG19 layers to tailor the model for the image classification task. These supplementary layers include:

- a flattened layer to transform the outputs into a one-dimensional vector;
- dense (fully connected) layers for final classification, including dropout layers for regularisation;
- batch normalisation layers for normalising activations and stabilising learning;
- the final dense layer, with neurons corresponding to the number of classes (letters) in the classification task.

In our implementation, the input image size is (64,64,3), and the feature extraction part strictly follows the VGG19 configuration. It consists of five blocks of convolutional layers with small 3x3 kernels and a stride of 1. After flattening the output of the final block, we added two dense (fully connected) layers with 512 units each. Both are activated using ReLU and followed by a Dropout layer with a rate of 0.5 to reduce overfitting. Each dense block also includes Batch Normalisation to stabilise learning and improve convergence. The final output layer contains 26 neurons (one per ASL letter), activated using Softmax. The network was trained using the Adam optimiser with a $10^{-4}$ learning rate and categorical cross-entropy loss. Fig 3 shows the full architecture of the proposed model in a horizontal layout for clarity and reproducibility.

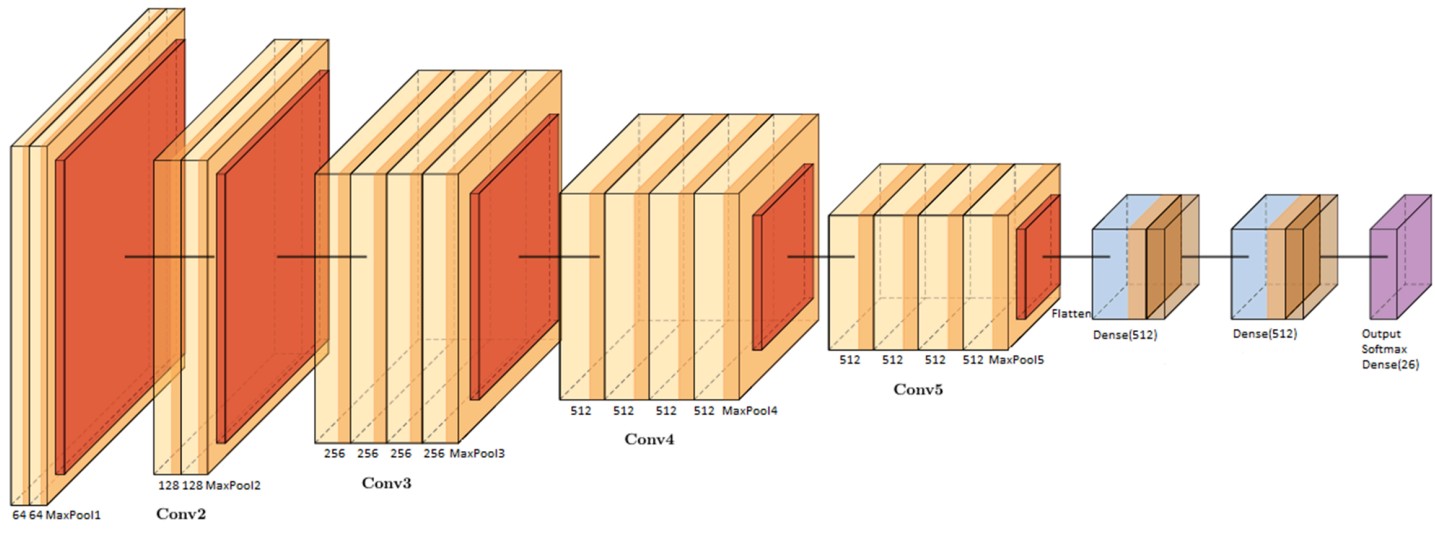

**Fig 3**. **Architecture of the proposed CNN model based on a modified VGG19 structure.** The network consists of five convolutional blocks (Conv1–Conv5), each followed by ReLU activations and max-pooling operations. After the feature extraction phase, the feature maps are flattened and passed through two fully connected (Dense) layers of 512 units each, followed by ReLU activations, Dropout (rate = 0.5), and Batch Normalisation. The final output layer uses a Softmax activation with 26 neurons to classify ASL alphabet signs.

## Model limitations and motivation for improvement

The results obtained show a test accuracy of 95%; however, when evaluated on out-of-distribution images, not present in the original dataset, the model exhibits a clear divergence in performance (https://www.kaggle.com/code/damionjoyner/sign-language-classification-cnn-vgg19). It fails even to recognise simple hand signs, revealing significant limitations in its generalisation ability. A more in-depth analysis of the model behaviour uncovers several fundamental weaknesses that severely undermine its capacity for real-world deployment. The training dataset is imbalanced in class distribution and limited in subject diversity, as it includes samples from only five individuals, all with similar skin tones. This dual constraint introduces a strong learning bias: the model tends to overfit to frequent classes and person-specific features, rather than learning gesture-invariant representations. As a result, its performance drops notably for rare classes and when applied to new users not represented in the training set. Furthermore, the small size of the dataset, combined with its imbalance, significantly increases the risk of overfitting. Although the overall accuracy appears high, this metric can be misleading. The model may simply memorise dataset-specific patterns rather than learning transferable features. This concern is reinforced by the inconsistencies observed during additional evaluations, particularly on underrepresented classes. Another critical limitation lies in the way multimodal information is handled. Although the model uses both RGB and depth data, the fusion is performed through simple concatenation, which does not fully leverage the complementary strengths of these modalities. This naive fusion strategy may cause the model to overlook essential modality-specific cues or to resolve ambiguities inadequately, shortcomings that more advanced fusion mechanisms could potentially overcome. These interconnected limitations expose the structural weaknesses of the current approach and motivate the need for methodological improvements. Specifically, they justify the development of a more robust architecture that employs advanced multimodal fusion techniques, applies targeted data augmentation to mitigate class imbalance, and incorporates regularisation strategies to promote generalisation. These enhancements form the basis of the improvements proposed in the following section.

## An enhanced interpretable solution for SLR

To overcome the limitations of existing ASL datasets and improve the robustness and interpretability of gesture recognition models, we propose a three-step approach: (1) data augmentation and diversification, (2) a mask-based preprocessing strategy, and (3) an explainable classification framework. These three steps are described in the following subsections.

**Data augmentation and diversification.** The accuracy of ASL recognition models is often hindered by the limited variability in existing datasets, particularly with respect to hand shape, skin tone, background, and lighting conditions. To address this, we significantly changed and diversified our dataset. Instead of relying on a standard collection of 100,000 images covering 24 letters, we compiled a new dataset of 325,000 cropped RGB images representing all 26 letters of the ASL alphabet as illustrated in Fig 4.

These images were gathered from four publicly available sources to ensure heterogeneity and cover a wide range of environments and hand configurations:

- Kaggle - ASL Alphabet https://www.kaggle.com/datasets/grassknoted/asl-alphabet
- Kaggle - ASL Alphabet Dataset https://www.kaggle.com/datasets/debashishsau/aslamerican-sign-language-aplhabet-dataset

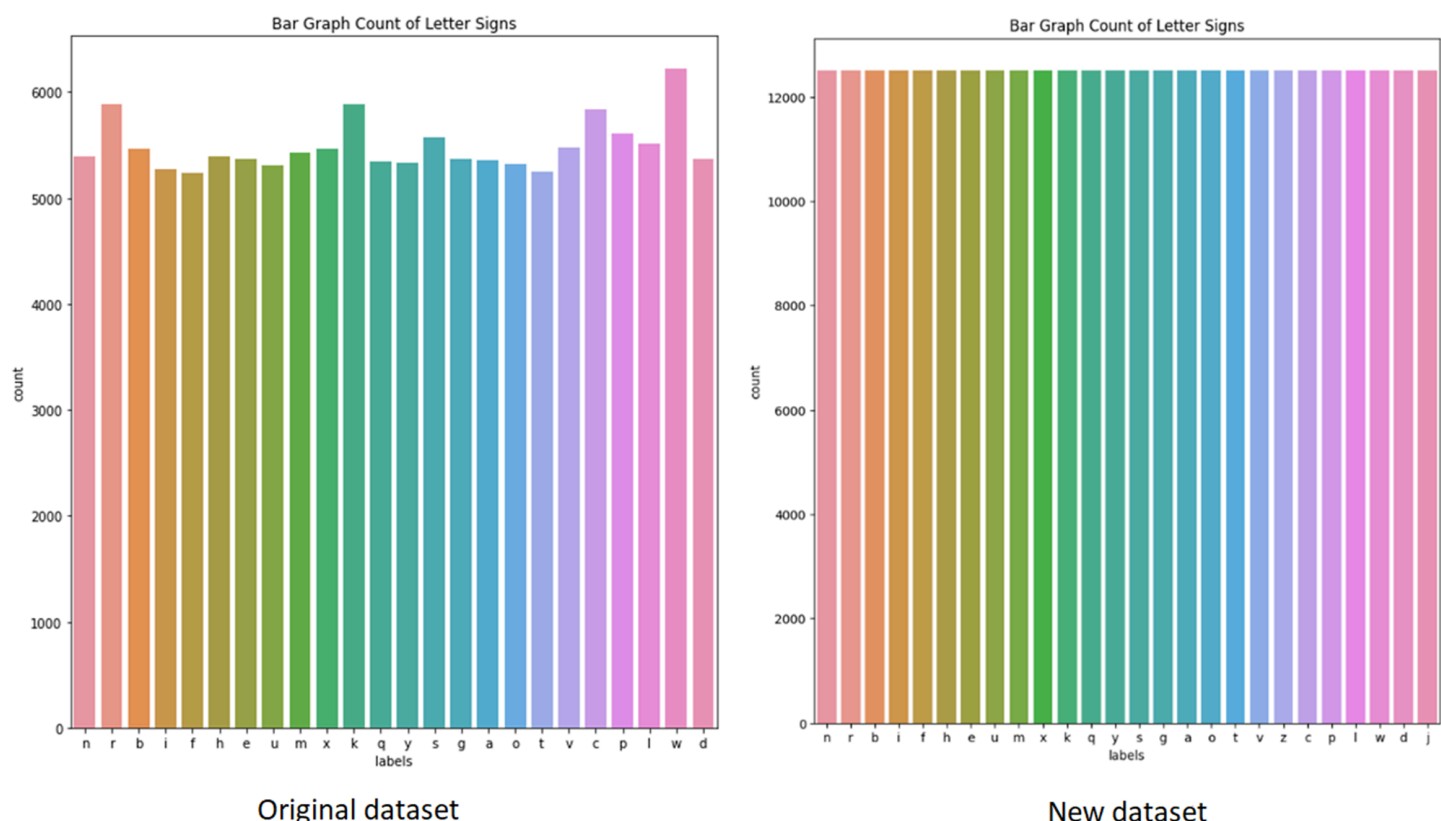

**Fig 4. Comparison of label distributions between the initial and the newly collected ASL letter datasets.** The left bar chart shows the original dataset with imbalanced counts across 24 letters, while the right chart illustrates the new dataset, which includes 12,500 per letter and covers all 26 letters of the ASL alphabet with uniform distribution.

- Kaggle - ASL Alphabet Test https://www.kaggle.com/datasets/danrasband/asl-alphabet-test
- Kaggle - Synthetic ASL Alphabet https://www.kaggle.com/datasets/lexset/synthetic-asl-alphabet

The dataset was constructed by merging multiple publicly available ASL alphabet datasets. After removing duplicates and balancing the classes, we obtained approximately 325,000 images (12,500 per class). Since the images were mixed and reorganized into a unified dataset, we report only the final distribution. The processed dataset and split files are made publicly available at (https://www.kaggle.com/datasets/fatimaelqoraychy/dataset). Although 'j' and 'z' are dynamic ASL signs requiring motion to be fully expressed, some of the datasets we employed include static image approximations for these letters. In this study, we used these static representations to cover the complete 26-letter ASL alphabet. While these approximations provide useful proxies, they cannot fully reproduce the motion aspect of the signs. In this study, we did not employ geometric or photometric data augmentation (e.g., rotations, shifts, zooms, flips, or brightness variations). The only preprocessing applied was pixel-value rescaling (1/255), performed online during training using the Keras `Image-DataGenerator`. This ensured consistent normalization across the dataset without altering the original distribution of hand images.

**Segmentation-based preprocessing with U-Net.** Previous modelling methods often rely on depth images to capture 3D hand structures. However, depth data is highly sensitive to background clutter, lighting inconsistencies, and camera angle variations, making it less reliable for robust gesture recognition. To overcome these challenges, we propose a segmentation-based preprocessing approach that focuses on binary hand masks instead of depth images. We employed the U-Net architecture [52] for this task. U-Net is a convolutional neural network designed for precise image segmentation. Its encoder-decoder structure, combined with skip connections, allows it to retain both high-level semantics and spatial resolution, making it particularly effective for extracting hand regions from complex scenes. To train this network and produce a segmentation model, we used the HGR1 dataset (https://sun.aei.polsl.pl// mkawulok/gestures/), which contains 899 labelled RGB images with corresponding binary masks. This dataset includes diverse hand poses captured under varying lighting conditions, backgrounds, and camera types, making it suitable for training a robust and generalizable segmentation model. The resulting binary masks isolate the hand region, removing irrelevant background information. This facilitates a more focused and noise-free input for classification, allowing the model to better capture critical gesture features such as finger positions and hand contours.

The segmentation module is implemented as a 5-level U-Net with skip connections designed to predict binary hand masks from 128×128×3 RGB inputs. Each encoder stage consists of two 3×3 convolutional layers with ReLU activation and "same" padding, followed by 2×2 max pooling for downsampling. The number of feature channels increases with depth, following the sequence $\{16, 32, 64, 128, 256\}$. The decoder mirrors this structure: at each stage, a 2×2 up-sampling layer is concatenated with the corresponding encoder features, and followed by two 3×3 Conv–ReLU layers. A final 1×1 convolution with `sigmoid` activation produces a per-pixel probability map of the hand region. The network is optimized with RMSprop and trained with mean squared error (MSE) loss for 50 epochs using a batch size of 32. All input images are normalized to the [0,1] range. To ensure fair evaluation and prevent identity leakage, we apply a subject-disjoint split for train, validation, and test sets, and monitor validation performance during training. The predicted hand masks serve two purposes within the proposed framework. First, they constitute the input to the mask-based classifier, enabling the comparison between RGB and silhouette representations. Second, they support the quantitative explainability analysis by providing a reference structure to verify whether attribution maps align with the hand regions. Representative qualitative results, including input images, predicted masks, and overlays, are reported to illustrate both typical successes and failure cases.

**Model training and evaluation.** Following the segmentation process, two distinct gesture recognition models were trained. The first model was trained on the original RGB images, while the second model utilised the binary hand masks generated by the U-Net segmentation model. The dataset collection and processing workflow began with acquiring a total of 325,000 labelled images. Each image underwent two distinct preprocessing stages: first, it was used as an RGB

image in its original form, and second, it was passed through the segmentation model to generate the corresponding hand masks. All images were resized to a fixed resolution of 64×64 pixels, which is the required input dimension for the VGG19-based architecture used in our models as presented in Fig 5. To enable reliable model evaluation, the dataset was split into 76.5% for training, 13.5% for validation, and 10% for testing, using random shuffling and a fixed random seed labeled SEED equal to 99 to ensure reproducibility. This guarantees consistent data partitioning across different runs. To monitor and mitigate overfitting during training, we recorded both training and validation losses at each epoch across all 25 training epochs. Early stopping was applied by monitoring the validation loss. The training was halted using an early stopping criterion once no improvement was detected over a set number of consecutive epochs. The test set was used to evaluate the final model after training was complete. A summary of the training hyperparameters is presented in Table 2. All experiments were conducted using the Kaggle cloud-based environment, which provides NVIDIA Tesla P100 GPUs. Each model was trained for approximately six hours.

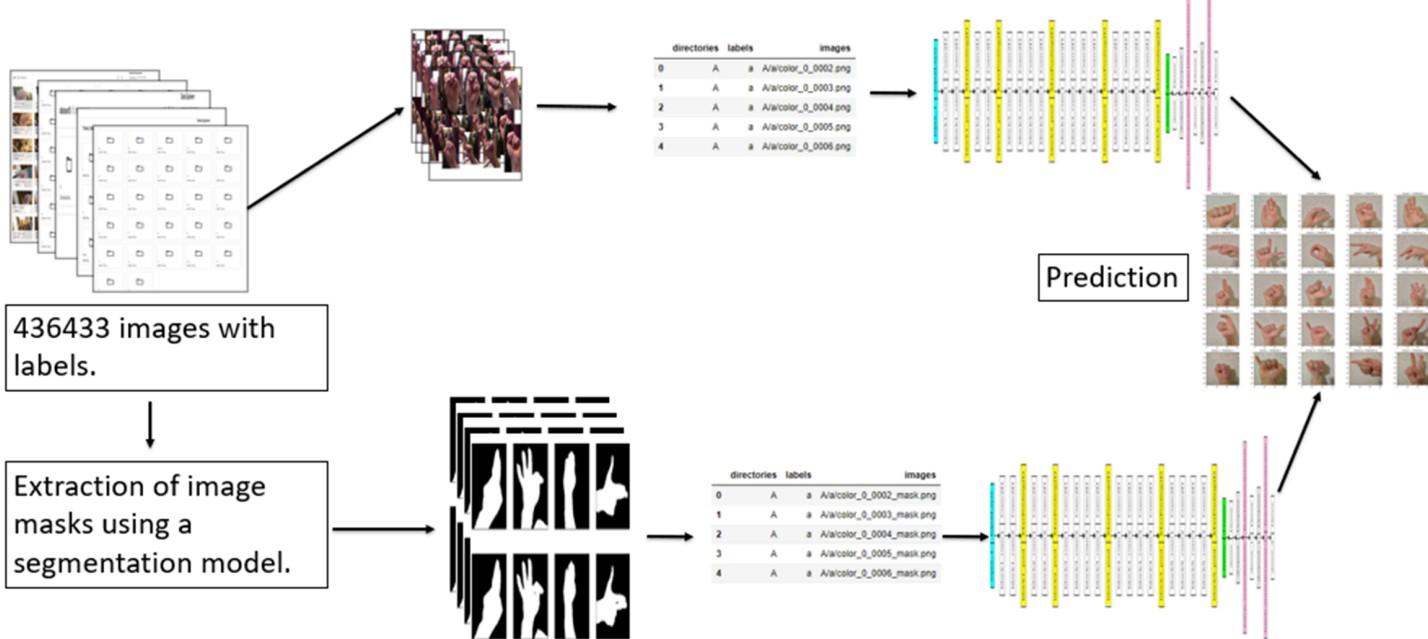

**Fig 5**. **Overview of the gesture classification process using RGB images and segmented hand masks.** Both input types are preprocessed and passed through a CNN for gesture prediction.

**Table 2**. **Hyperparameters used in training the classification models.**

| Hyperparameter | Value |
|---|---|
| Input size | 64×64×3 |
| Batch size | 64 |
| Optimizer | Adam |
| Learning rate | 0.0001 |
| Dropout rate | 0.5 |
| Activation (hidden layers) | ReLU |
| Activation (output) | Softmax |
| Loss function | Categorical cross-entropy |
| Epochs | 25 |

### Real-time prediction interface

We developed a real-time prediction interface using OpenCV and TensorFlow to evaluate model performance in real-world settings. The interface captures live images of hand gestures through a webcam and performs on-the-fly predictions using the pre-trained model. It simultaneously displays the predicted ASL letter and overlays visual explanations generated through integrated XAI methods.

The interface operates by continuously capturing frames, resizing them to 64×64 pixels to match the model's input dimensions, and normalising the pixel values. The processed frame is then passed through the model to obtain prediction probabilities, and the class with the highest score is selected as the predicted letter. The predicted class index is converted back into a human-readable label using a loaded label binarized object.

**Performance evaluation.** To quantify the real-time usability of the system, we measured both frame rate (frames per second, FPS) and per-frame latency (milliseconds). On our test hardware (Intel CPU), the interface achieved an average of 10 FPS with a mean latency of approximately 100 ms per frame.

### Explainable AI for SLR model enhancement

We apply XAI techniques to our SLR models to enhance transparency and interpretability. In ASL recognition, it is crucial to ensure that a model focuses on semantically meaningful regions of the input, such as hand shapes and orientations, rather than background artefacts. This is particularly important for model validation and for fostering confidence among users and system developers. Explainability also serves as a practical tool for model debugging and for detecting potential biases or spurious correlations learned during training [44]. For instance, if a model consistently misclassifies a gesture due to attention on irrelevant background elements, explainability methods can help identify and correct such failures. Furthermore, it provides traceability for misclassifications by highlighting decision-relevant input regions. However, due to ASL gestures' complexity and spatial variability, no single XAI technique is sufficient. We therefore employ three complementary and widely recognised XAI methods: Grad-CAM, LIME, and integrated gradients.

**Gradient-weighted class activation mapping.** Gradient-weighted Class Activation Mapping (Grad-CAM) [23] is a widely used post-hoc explainability technique that provides visual explanations for CNNs by highlighting image regions most relevant to a model's decision. Given an input image and a target class $c$, Grad-CAM produces a heatmap that localises the spatial regions that have the strongest influence on the output score $y^c$. This is particularly useful in visual tasks like SLR, where fine-grained hand features drive classification. To compute Grad-CAM, we first calculate the gradient of the output score $y^c$ concerning the feature maps $A^k$ of a chosen convolutional layer:

$$\frac{\partial y^c}{\partial A_{ij}^k}$$

These gradients are globally averaged to obtain importance weights $\alpha_k^c$ for each feature map:

$$\alpha_k^c = \frac{1}{Z} \sum_i \sum_j \frac{\partial y^c}{\partial A_{ij}^k}$$

Where $Z$ is the total number of pixels in the feature map.

The class-discriminative localization map $L_{\text{Grad-CAM}}^c$ is then computed as:

$$L_{\text{Grad-CAM}}^c = \text{ReLU}\left(\sum_k \alpha_k^c A^k\right)$$

The ReLU operation retains only features with a positive influence on the class score. The resulting heatmap is then upsampled and overlaid on the input image using a JET colourmap, where red and yellow indicate high importance and blue indicates low relevance.

**Local interpretable model-agnostic explanations.** Local Interpretable Model-agnostic Explanations (LIME) [16] is a widely used technique designed to interpret the predictions of any black-box model by approximating it locally with an interpretable model. Unlike Grad-CAM, which is specific to CNNs, LIME is model-agnostic and can be applied to any classifier. The core idea is to understand a model's decision for a specific instance $x$ by generating a set of perturbed samples around $x$ and observing how the model's predictions vary. For each perturbed sample $x'$, the model predicts an output $f(x')$, and a locality-aware weight $\pi_x(x')$ is assigned based on the proximity to the original instance $x$. A simple, interpretable model $g$, such as a linear regression or decision tree, is trained on this locally weighted dataset to approximate the complex model $f$ in the vicinity of $x$. The optimisation objective is:

$$\arg\min_{g \in G} \mathcal{L}(f, g, \pi_x) + \Omega(g)$$

where $\mathcal{L}$ measures the fidelity of $g$ to $f$ locally, $\pi_x$ defines the locality, and $\Omega(g)$ is a complexity penalty for $g$. In image classification, LIME segments the image into super pixels and perturbs it by randomly turning these super pixels on or off. It then identifies which super pixels most strongly influence the prediction by observing the model's output across these perturbations.

**Integrated gradients.** Integrated Gradients [53] is a model-specific explainability method designed to attribute a deep neural network's prediction to its input features. It addresses the limitations of standard gradients, which can be noisy or vanish due to saturation. The method works by computing the average gradients as the input transitions from a baseline (typically a zero or black image) to the actual input. For an input $x$ and baseline $x'$, the attribution for the $i$-th feature is defined as:

$$IG_i(x) = (x_i - x_i') \times \int_{\alpha=0}^{1} \frac{\partial f(x' + \alpha(x - x'))}{\partial x_i} \, d\alpha$$

This integral is approximated numerically via summation over $m$ steps:

$$IG_i(x) \approx (x_i - x_i') \times \frac{1}{m} \sum_{k=1}^{m} \frac{\partial f\left(x' + \frac{k}{m}(x - x')\right)}{\partial x_i}$$

The integrated gradients method satisfies desirable axioms such as sensitivity and implementation invariance, making it theoretically sound for attribution tasks.

## Evaluation metrics for saliency maps

To quantitatively assess the compactness and dispersion of the explanations, We computed two complementary metrics on the saliency maps produced by Grad-CAM, LIME, and Integrated Gradients.

**Energy concentration.** Let $H \in \mathbb{R}^{m \times n}$ denote the normalized saliency map, with $H_{ij} \geq 0$ and $\sum_{i=1}^{m} \sum_{j=1}^{n} H_{ij} = 1$. After sorting all pixel intensities in descending order $h_{(1)} \geq h_{(2)} \geq \cdots \geq h_{(mn)}$, the energy concentration at fraction $p$ (e.g., $p = 0.05$ for top-5%) is defined as:

$$E(p) = \frac{\sum_{i=1}^{\lfloor p \cdot mn \rfloor} h_{(i)}}{\sum_{i=1}^{mn} h_{(i)}}. \tag{1}$$

A higher $E(p)$ indicates that a small fraction of pixels captures most of the attribution energy, i.e., the explanation is more compact and localized.

**Entropy.** Considering the normalized heatmap as a discrete probability distribution, The entropy is defined as:

$$\mathcal{H}(H) = -\sum_{i=1}^{m}\sum_{j=1}^{n} H_{ij} \log\left(H_{ij} + \epsilon\right), \tag{2}$$

where $\epsilon$ is a small constant to prevent numerical instability. Lower entropy values correspond to more focused saliency maps, while higher values reflect more diffuse explanations.

### A multi-level interpretability strategy for SLR

The decision to integrate multiple XAI techniques, rather than rely on a single method, is motivated by the complementary nature of the insights each one provides. Grad-CAM, LIME, and integrated gradients represent distinct families of interpretability methods: Grad-CAM leverages convolutional activations to visualise class-specific saliency maps, LIME builds local surrogate models through feature perturbation, and integrated gradients quantify the cumulative contribution of each input pixel via a path-integrated gradient from a baseline. Each method brings specific strengths and limitations. Grad-CAM produces intuitive heatmaps but may lack fine-grained spatial resolution. LIME offers detailed, instance-level attributions but can be unstable across similar inputs. The integrated gradients method provides pixel-level precision but is sensitive to baseline selection and may miss high-level spatial structure. By combining all three, we obtain a multi-level perspective of the model's internal logic—from broad spatial focus to localised feature attribution and cumulative influence. This combined framework is particularly crucial in SLR, where classification hinges on nuanced finger position, contour, and orientation differences. Relying on a single method may result in incomplete or misleading interpretations. Therefore, the integration of Grad-CAM, LIME, and integrated gradient methods is not redundant—it is essential for achieving robust, trustworthy interpretability, enabling the diagnosis of both successful and failed predictions with higher fidelity.

For quantitative evaluation, we report precision, recall, and F1-score. These metrics are particularly appropriate for our context due to class imbalance and the presence of visually ambiguous gestures. Precision measures the exactness of the predictions, recall assesses completeness, and F1-score offers a harmonic balance of both. Unlike overall accuracy, these metrics provide a more nuanced assessment of model behaviour, especially on minority or error-prone classes. To guide both interpretability and performance analysis, we selected representative subsets of ASL letters:

- **Simple Signs:** 'a' and 'l'
- **Challenging Pairs:** 'i' vs. 'j', and 'm' vs. 'n'

The first pair ('a' and 'l') serves as a baseline to verify that the model correctly handles straightforward gestures. The second and third sets represent more ambiguous cases with high visual similarity. These allow us to test the model's discriminative ability and examine how explanation methods respond in both correct and incorrect classifications under challenging conditions. All scripts and code used to generate the results in this study are available on Kaggle at https://www.kaggle.com/code/fatimaelqoraychy/sign-language-classification-cnn-vgg19-1. The notebook is publicly accessible and can be used to reproduce the experiments reported in this manuscript.

### Results

This section introduces the results of the empirical work described in the previous sections. Such results are grouped into two parts: one related to the model's accuracy and one related to its explainability.

## Model for gesture recognition

As shown in Figs 6 and 7, both models converge early during training. The RGB model reached optimal performance at epoch 13, while the mask-based model continued improving until epoch 24. This difference reflects the RGB model's faster learning due to richer visual features, whereas the mask model, relying solely on shape information, required more epochs to generalise effectively. In both cases, the use of early stopping ensured stable convergence and prevented overfitting. Both models demonstrated high performance in recognising ASL gestures during the test, with strong average precision, recall, and F1-scores. While Table 4 provides an overview of global performance, a more detailed breakdown is presented in Table 3, which reports the exact values of each metric for every letter in the alphabet.

To evaluate model performance in a real-world scenario, we implemented a real-time prediction interface capable of recognising ASL letters as presented in Sect . During this phase, we observed that the RGB model consistently predicted the correct letter, even for complex signs. However, in the case of 'i' and 'j', despite their similarity, the model provided the correct prediction. To better understand the source of these errors, we applied explainability methods to analyse which image regions the model relied on during prediction.

## Explainability and model validation

To better understand our system's prediction mechanisms, we analyze visual explanations generated by Grad-CAM, LIME, and integrated gradient methods across the two classification models. While the RGB model achieves consistently high accuracy across all letters, the mask-based model occasionally struggles, particularly confusing visually similar signs such as 'n' and 'm'. Interestingly, it correctly classifies 'm' and successfully distinguishes other subtle variations such as 'i' and 'j'. To systematically investigate this behaviour, we selected representative signs for qualitative analysis: simple and clearly distinguishable gestures ('a', 'l'), a visually similar but correctly classified pair ('i', 'j'), and a confused pair ('m', 'n'). This strategy allows us to assess what features the model learns, how it fails, and how explanations differ between RGB and mask-based models.

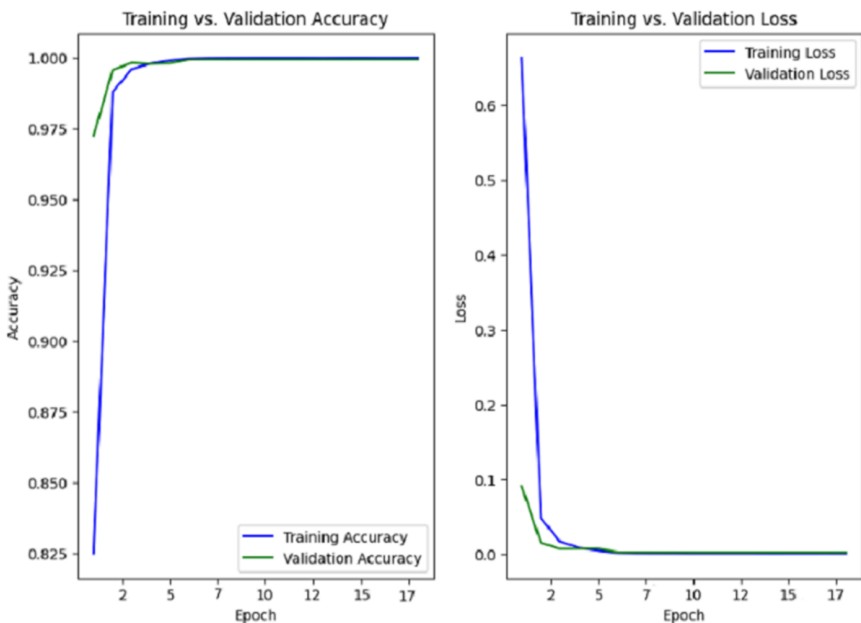

**Fig 6**. **Training and validation accuracy and loss curves for the RGB model.**

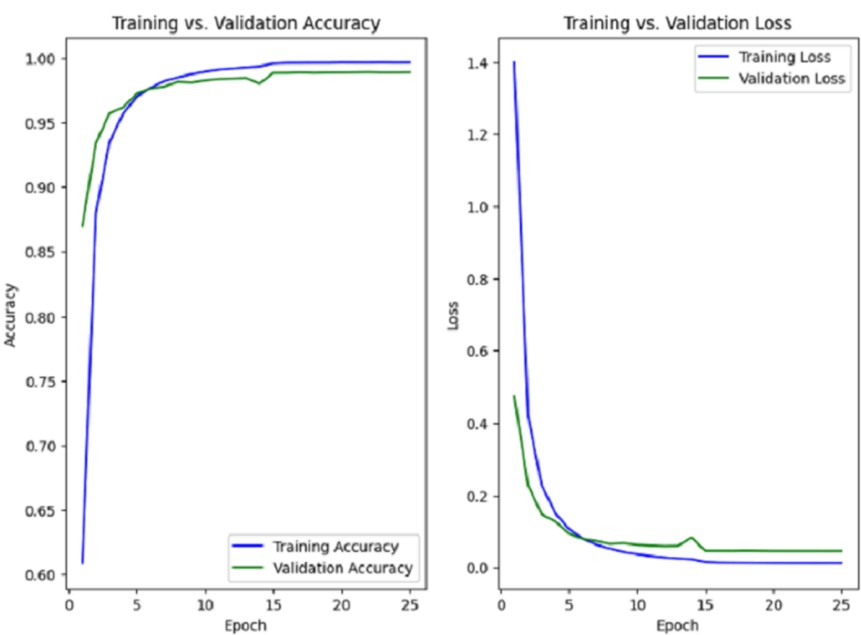

**Fig 7**. **Training and validation accuracy and loss curves for the Mask-based model.**

**Table 3**. **Classification of ASL letters in the testing phase and comparison of classification performance (Precision, Recall, F1-score) between the RGB and Mask models.**

| Letter | Precision | | Recall | | F1-score | |
|---|---|---|---|---|---|---|
| | **RGB** | **Mask** | **RGB** | **Mask** | **RGB** | **Mask** |
| a | 1.00 | 0.97 | 0.99 | 0.98 | 1.00 | 0.97 |
| b | 1.00 | 0.98 | 1.00 | 0.98 | 1.00 | 0.99 |
| c | 0.99 | 0.99 | 1.00 | 0.99 | 0.99 | 0.98 |
| d | 1.00 | 0.97 | 0.99 | 0.99 | 1.00 | 0.98 |
| e | 1.00 | 0.98 | 0.98 | 0.98 | 0.99 | 0.98 |
| f | 1.00 | 0.98 | 1.00 | 0.98 | 1.00 | 0.98 |
| g | 0.99 | 0.98 | 0.99 | 0.98 | 0.99 | 0.98 |
| h | 0.99 | 0.98 | 1.00 | 0.98 | 1.00 | 0.98 |
| i | 1.00 | 0.99 | 1.00 | 0.98 | 1.00 | 0.98 |
| j | 1.00 | 1.00 | 1.00 | 0.98 | 1.00 | 0.99 |
| k | 1.00 | 0.99 | 0.99 | 0.99 | 0.99 | 0.99 |
| l | 1.00 | 0.99 | 1.00 | 0.99 | 1.00 | 0.99 |
| m | 1.00 | 0.98 | 0.97 | 0.96 | 0.98 | 0.97 |
| n | 0.97 | 0.96 | 0.99 | 0.95 | 0.98 | 0.96 |
| o | 1.00 | 0.98 | 1.00 | 0.96 | 1.00 | 0.97 |
| p | 1.00 | 0.91 | 1.00 | 1.00 | 1.00 | 0.95 |
| q | 1.00 | 0.97 | 1.00 | 0.98 | 1.00 | 0.98 |
| r | 0.99 | 0.98 | 1.00 | 0.97 | 0.99 | 0.98 |
| s | 0.98 | 0.98 | 1.00 | 0.97 | 0.99 | 0.98 |
| t | 1.00 | 0.97 | 0.99 | 0.96 | 1.00 | 0.96 |
| u | 1.00 | 0.97 | 0.99 | 0.96 | 0.99 | 0.96 |
| v | 1.00 | 0.97 | 1.00 | 0.96 | 1.00 | 0.96 |
| w | 1.00 | 0.96 | 1.00 | 0.96 | 1.00 | 0.96 |
| x | 0.99 | 0.97 | 1.00 | 0.98 | 0.99 | 0.97 |
| y | 1.00 | 0.97 | 1.00 | 0.98 | 1.00 | 0.98 |
| z | 1.00 | 1.00 | 1.00 | 1.00 | 1.00 | 1.00 |

**Table 4**. Performance comparison between the RGB and Mask models for ASL gesture recognition.

| Model | Precision | Recall | F1-Score |
|---|---|---|---|
| RGB Model | 0.99 | 0.99 | 0.99 |
| Mask Model | 0.98 | 0.98 | 0.98 |

**Simple Signs 'a' and 'l':**  For the simple letters 'a' and 'l', all three methods show strong alignment between the model's focus and the relevant gesture areas, as illustrated in Fig 8. Grad-CAM consistently highlights the hand's central region, capturing the gesture's overall shape, particularly the closed fist for 'a' and the raised thumb and index for 'l'. This suggests the model is using spatially relevant information to make correct predictions. LIME provides contour-based insights, outlining the hand boundaries precisely. The highlighted regions correspond well with the hand's silhouette, indicating that the model is sensitive to the global shape and the number of extended fingers. The integrated gradients

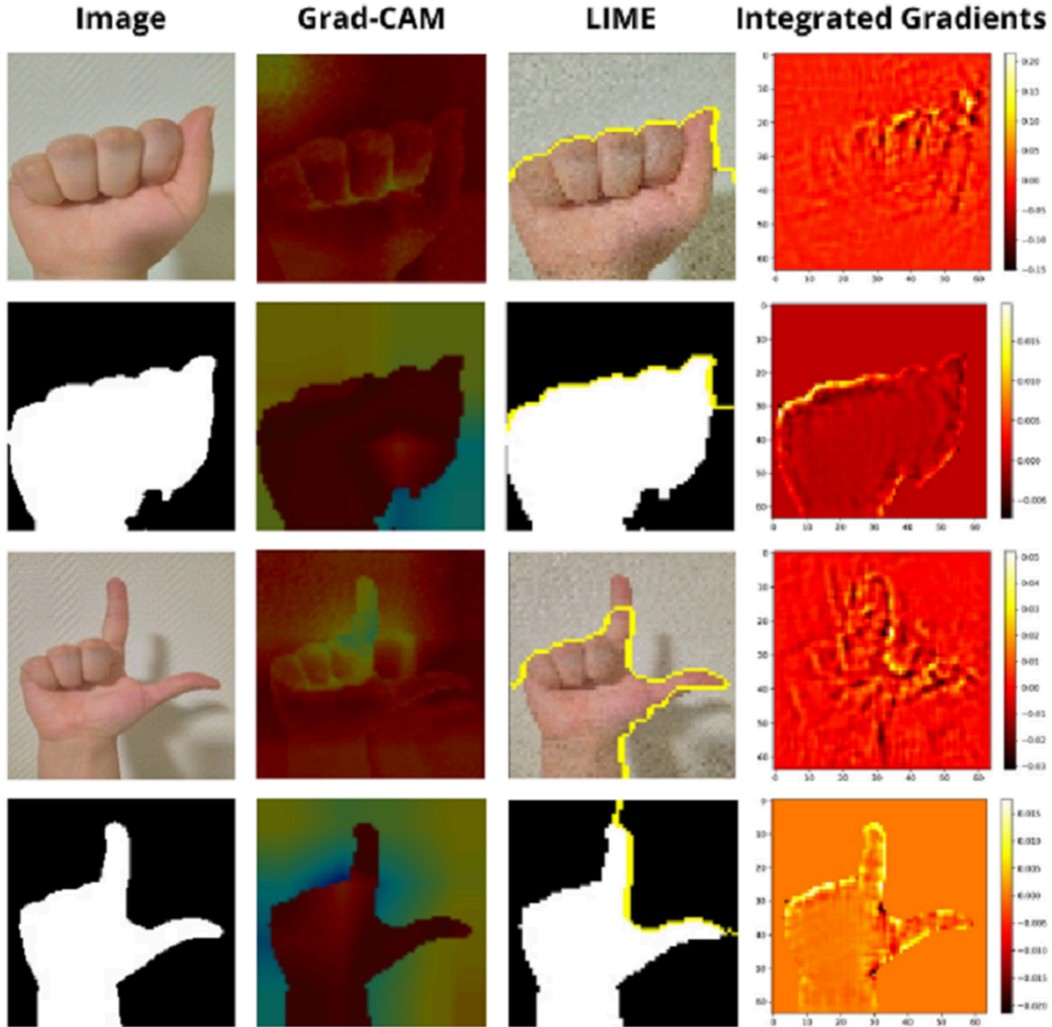

**Fig 8**. **Explainability results for the simple ASL letters 'a' and 'l' using Grad-CAM, LIME, and integrated gradients methods.** Each column corresponds to a different explanation technique, while each row shows original inputs and their corresponding binary masks.

method shows high activation in areas with clear structural edges, such as knuckles, fingertips, and wrists. These results confirm that pixel-level contributions are coherent with human intuition. This consistency across methods confirms the model's robustness in recognising simple, well-separated gestures. These signs serve as effective baselines for validating that the explainability tools reflect correct model behaviour.

**Challenging Signs 'i' vs. 'j':** The distinction between i' and j' lies primarily in the little finger's slight curvature or motion-related aspect. Despite their visual similarity, both the RGB and mask-based models correctly classify these signs (Fig 9). Grad-CAM tends to produce similar heatmaps for both letters, often centred on the palm and lower fingers, suggesting reliance on coarse spatial features. LIME adds nuance while 'i has tightly bound contours around the little, 'j' frequently shows extended contours, sometimes trailing into the curved stroke direction, hinting that LIME can capture implied motion. The integrated gradients method exposes localised pixel importance, with 'j' often showing extended activation beyond the little, consistent with its curved nature. The combined analysis suggests that while Grad-CAM alone may miss subtle motion cues, LIME and integrated gradients provide additional evidence that the model captures fine variations in finger trajectory.

**Challenging Signs 'm' vs. 'n':** The pair 'm' and 'n' presents a fine-grained classification challenge, both involve fingers folded over the thumb, differing primarily in the number of visible knuckles—three for 'm' and two for 'n'. While the RGB model correctly classifies both signs, the segmentation-based model systematically misclassified 'n' as 'm', as shown in Fig 9. Grad-CAM shows that in the mask-based model, both 'm' and 'n' result in nearly identical heatmaps, with broad activation spread over the entire hand region. This suggests that the model fails to localise the discriminative area, the number of folded fingers, which is critical for distinguishing the two letters. In contrast, the RGB model shows slightly more concentrated activation toward the finger area for 'm', reflecting finer spatial awareness. LIME provides a more informative contrast. For the RGB model, LIME outlines distinct superpixels around three folded fingers for 'm' and two for 'n', helping the model separate the two signs. However, in the mask-based model, the superpixel boundaries for 'n' are either missing, merged, or appear very similar to those for 'm'. This visual similarity may explain why the model maps to the same internal representation. The integrated gradients method confirms this limitation. In the RGB model, the attribution maps for 'm' and 'n' highlight different finger segments with clear separation, three highlighted areas for 'm', two for 'n'. However, in the mask-based model, both maps show overlapping attribution across the top of the hand, with weak contrast between the two and three finger zones. This blurs the structural boundary between the two classes. These results suggest that the mask model fails to attend to the correct discriminative region (i.e., finger count), likely due to the absence of texture, shading, and depth cues. Without this fine-grained information, the visual difference between 'm' and 'n' collapses, and the model overgeneralizes, classifying both as 'm'. In contrast, the RGB model benefits from additional visual signals that help preserve subtle structural differences.

To quantitatively assess the compactness and dispersion of the explanations, we computed two complementary metrics on the saliency maps produced by Grad-CAM, LIME, and Integrated Gradients. Energy concentration was defined as the proportion of attribution energy contained in the top $k$% most salient pixels ($k = 5, 10, 20$). Higher values indicate that the explanation is concentrated on fewer, more relevant regions. Entropy was calculated over the normalized heatmap to measure dispersion. Lower entropy values correspond to more focused explanations, while higher values indicate more diffuse attention. These metrics were computed for six representative ASL letters (*a, l, i, j, m, n*) and across both models (RGB- and mask-based), then averaged to provide a global comparison. Table 5 summarizes the averaged results across all letters. The results reveal distinct behaviors across the three explanation techniques. **LIME** produces the most compact explanations, with the highest concentration values (e.g., 21.6% at top-10%) and the lowest entropy (7.9), indicating highly localized attribution maps. **Grad-CAM**, by contrast, exhibits more diffuse saliency maps, reflected in the highest entropy (9.3) and only moderate energy concentration (13.5% at top-10%). **Integrated Gradients** yields the lowest

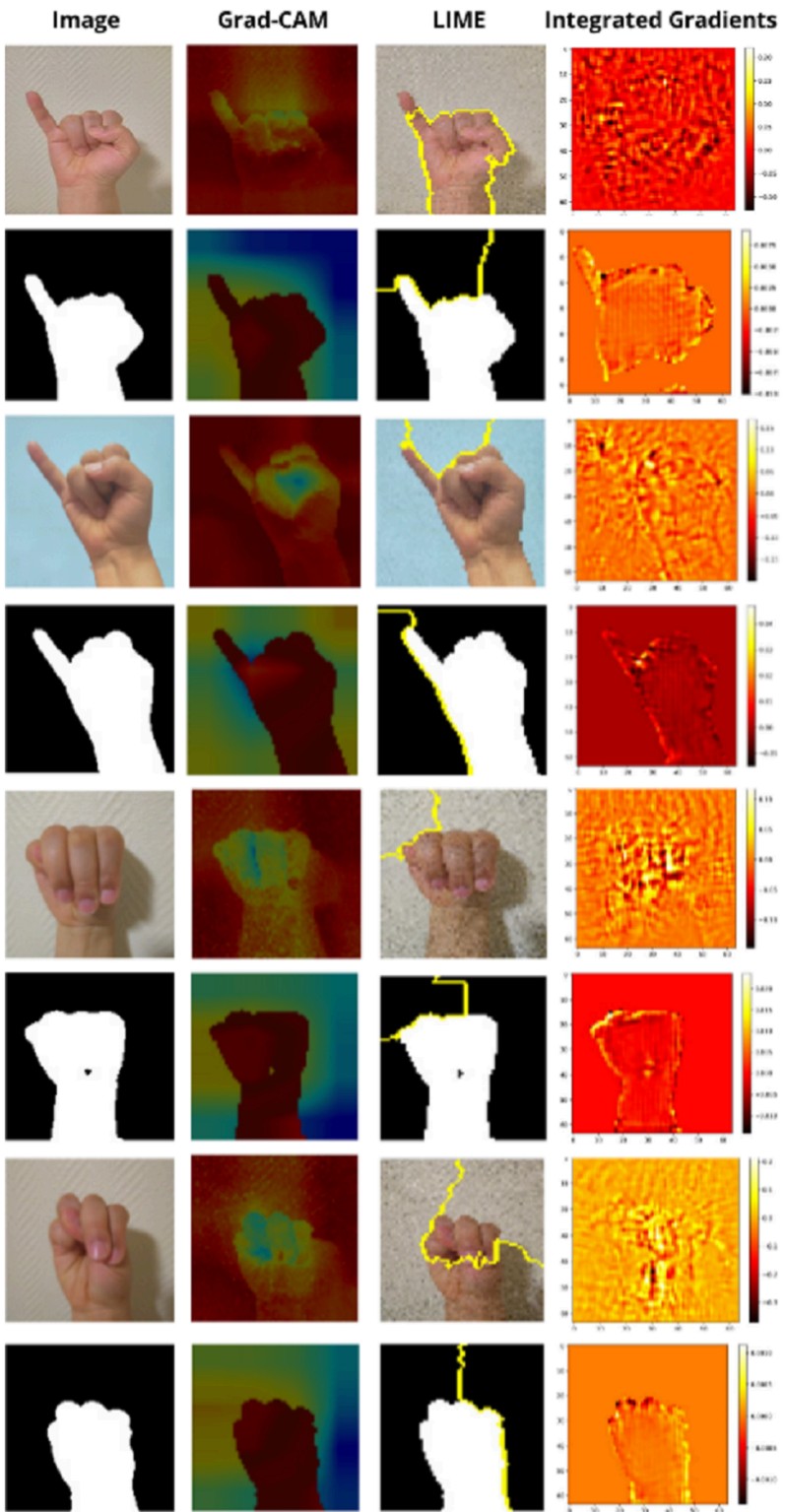

**Fig 9. Explainability analysis for visually similar ASL signs: 'i' vs. 'j' and 'm' vs. 'n' using Grad-CAM, LIME, and integrated gradients methods.** Each column corresponds to a different explanation technique, while each row shows original inputs and their corresponding binary masks.

**Table 5. Average energy concentration (computed at top-*k*% pixels) and entropy.** Values are averaged across six representative letters (*a, l, i, j, m, n*) and across both input modalities (RGB and mask). Higher energy concentration indicates more compact explanations; lower entropy indicates less diffuse maps.

| Method | Avg. Top-5% | Avg. Top-10% | Avg. Top-20% | Avg. Entropy |
|---|---|---|---|---|
| Grad-CAM | 7.3% | 13.5% | 29.5% | 9.32 |
| LIME | 10.8% | 21.6% | 43.1% | 7.91 |
| Integrated Gradients | 6.8% | 12.4% | 23.0% | 8.31 |

energy concentration overall (12.4% at top-10%), with entropy values (8.3) falling between LIME and Grad-CAM, suggesting smoother and more spread attributions. Overall, these results confirm that the three XAI methods provide complementary perspectives: LIME highlights precise local dependencies, Grad-CAM emphasizes broader spatial regions, and Integrated Gradients distributes importance across both local and contextual cues. This quantitative analysis strengthens the qualitative observations and underlines the benefit of using multiple explanation techniques for robust model auditing in sign language recognition.

## Implications for big data

Recent advances in sign language recognition highlight the increasing availability of large-scale datasets covering multiple sign languages, signing styles, and recording conditions. Such variability reflects a Big Data challenge, where models must generalize across highly heterogeneous samples. Our work is relevant in this context because it integrates explainability into SLR, providing tools to validate and interpret models when trained on large and diverse datasets. In particular, the integration of multiple XAI methods (Grad-CAM, LIME, and Integrated Gradients) facilitates model auditing at scale by enabling transparent validation of predictions, even in massive datasets where manual inspection of each sample is infeasible. Moreover, as highlighted in recent studies on the AI-powered evolution of big data [54], explainability is increasingly recognized as a key requirement for deploying trustworthy models in real-world, data-intensive environments. Therefore, we view our contribution as a step toward scalable and interpretable SLR systems capable of operating under Big Data conditions.

## Discussion

The experimental results presented in this study demonstrate high predictive performance for both RGB and mask-based models and expose deeper insights into the model behaviors through the lens of explainability. While the overall classification accuracy appears strong for both models (RGB: 99%, Mask: 98%), a deeper examination reveals key differences in generalisation and robustness. The RGB-based model consistently achieves better performance across ambiguous sign pairs, such as 'm' vs. 'n', which differ primarily by the number of extended fingers. This advantage stems from its access to rich visual cues, such as texture, shading, and skin tone, that enable the capture of subtle features. In contrast, despite eliminating background noise, the segmentation-based model lacks these visual depths and hence struggles with fine-grained discrimination. Moreover, the mask-based model demonstrates better robustness under noisy or cluttered environments, particularly in real-time settings where background conditions are variable. Its reliance on binary shape information makes it less sensitive to colour variations, lighting inconsistencies, or camera artefacts. This trade-off between robustness and discriminative power highlights the need for hybrid strategies that can balance visual richness with noise resilience.

We did not include a quantitative comparison with the baseline VGG19 model of Joyner, as this model failed to provide valid predictions on our datasets. This limitation was also reported by the original author on the project website, where it is noted that the model struggles with out-of-distribution samples. Our proposed models, by contrast, achieved consistent and interpretable results under the same conditions. One of this work's most salient contributions is the comparative

use of three XAI methods: Grad-CAM, LIME, and integrated gradients. Each explainability method brings a unique perspective: Grad-CAM identifies broader spatial attention, LIME reveals local sensitivity to super pixel-level features, and integrated gradients provide cumulative pixel-level attribution. While Grad-CAM offers intuitive heatmaps, it often lacks the resolution to distinguish fine-grained structural differences. LIME, though more precise, can vary between samples. The integrated gradients method excels at identifying localised saliency but struggles to convey global context. Their combined use triangulates the model's behaviour, helping mitigate the limitations of any one method. Results show that for simple and clearly separated gestures such as 'a' and 'l', all three methods align in highlighting the correct hand regions across both models. This consistency supports the model's ability to learn relevant features and provides confidence in its correct predictions. However, for visually similar gestures, particularly 'm' vs. 'n' divergence emerges. While the RGB model correctly classifies both signs with precise attribution maps (e.g., three vs. two finger contours), the mask-based model frequently misclassified 'n' as 'm'. Explainability analysis reveals the root cause: Grad-CAM fails to focus on the relevant discriminative region; LIME outlines are inconsistent or overlapping; and integrated gradients show blurred attribution across the fingers. These results suggest that the mask-based model lacks the resolution and richness to differentiate subtle anatomical differences due to its reliance solely on shape. In contrast, both models perform well for 'i' vs. 'j'. This is likely because the primary distinction, little finger curvature or implied motion, is still partially preserved in the binary mask. This highlights that not all structural differences are equally affected by the loss of RGB information; the ability to recognise dynamic or curvature-based features may remain intact in segmented representations, whereas subtle finger count distinctions do not. The RGB model outperforms the mask-based model not only in quantitative metrics but also in interpretability fidelity. The presence of colour, texture, and shading allows the RGB model to extract richer features and localise attention more effectively. These visual cues are especially crucial in gestures with fine-grained variations. Conversely, the binary mask strips away such information, which hinders the model's ability to resolve ambiguous patterns like those in 'm' and 'n'. This analysis underscores that segmentation-based input alone is insufficient for high-resolution gesture classification. To address this, several strategies are recommended:

- **Feature Enhancement:** Preprocessing techniques such as edge enhancement or skeletal overlays could improve the visibility of finger contours.
- **Hybrid Input Models:** Combining RGB and mask inputs could provide complementary benefits, shape consistency from masks, and detail richness from RGB.
- **Attention Mechanisms:** Using attention-based networks may help the model focus more selectively on discriminative regions.
- **Explainability Guided Retraining:** Insights from Grad-CAM or integrated gradients could guide targeted data augmentation or hard example mining during training.

In contrast to recent approaches such as SignExplainer [45], which primarily rely on LIME and SHAP to interpret feature contributions within ensemble classifiers or [48] introduces self-supervised learning with vision transformers and interprets decisions using SHAP, our work offers a multimodal, multi-technique explainability framework tailored to deep learning architectures. More recently, FedXAI-ISL [55] has proposed a federated and explainable framework for Indian Sign Language, and an explainable real-time SLR pipeline [56] has been introduced to address deployment at scale under big data conditions. While these contributions highlight the growing relevance of explainability in sign language recognition, most rely on a single explainability paradigm or are limited to specific learning setups.

By integrating Grad-CAM, LIME, and Integrated Gradients across RGB and segmentation-based models, we provide spatial, local, and structural insights beyond feature attribution and support deeper model auditing. Trust and transparency are critical in the context of assistive technology. An accurate yet opaque model is insufficient for user acceptance. The explainability framework presented here allows developers and users alike to understand model limitations, validate predictions, and adapt the system more effectively to real-world scenarios. A final implication is to integrate explainability into

the model design pipeline rather than treating it as a post-hoc tool. Future models could incorporate attention mechanisms or self-supervised learning schemes explicitly guided by XAI feedback. For instance, models could be penalised during training if their attention diverges from annotated hand regions, encouraging alignment between learned representations and interpretable features. A limitation of our approach is the treatment of dynamic ASL signs 'j' and 'z'. Since our models were trained on static images, we relied on dataset-provided approximations of these letters. While this enables classification of all 26 alphabet signs, the absence of true temporal modeling may affect recognition performance for such dynamic gestures. Future work will address this gap by incorporating video-based datasets and spatio-temporal architectures (e.g., 3D CNNs, temporal convolutional networks, or recurrent models).

## Conclusion

This study investigated the integration of XAI techniques into ASL recognition models to improve interpretability and transparency. Two distinct classification models were developed: one trained on RGB images and the other on binary hand masks obtained using a U-Net segmentation model. The need to enhance model interpretability and better understand the visual cues underlying prediction decisions motivated this study. We employed three complementary XAI methods to evaluate performance and decision-making mechanisms: Grad-CAM, LIME, and integrated gradients. Grad-CAM identifies broad spatial regions of interest, LIME captures local dependencies through input perturbation, and integrated gradients quantify the contribution of individual pixels. Together, these methods were chosen because they offer a multifaceted view of how each model interprets ASL gestures. Experimental results indicate that both models achieved strong classification performance, with the RGB-based model slightly outperforming the mask-based model, especially for visually similar hand gestures. Nevertheless, the mask-based model demonstrated a consistent focus on gesture-relevant regions by eliminating background noise, an advantage in real-world scenarios with variable lighting or clutter. A core contribution of this work lies in showing how XAI methods can elucidate internal model logic. The visualisations confirmed that the models generally focus on semantically meaningful hand regions and also exposed inconsistencies in handling ambiguous gestures (e.g., 'i' vs. 'j', 'm' vs. 'n'). These inconsistencies highlight regions where the model's attention drifts from semantically relevant features, underscoring the limitations of relying solely on accuracy metrics. Such findings have practical implications: they increase model transparency, support trust building in user-facing applications, and help identify avenues for improvement. For example, attention maps revealed that the segmentation model is more sensitive to hand pose variations, suggesting that incorporating pose normalisation or fusing RGB and mask data could improve robustness. Additionally, the models sometimes failed to distinguish dynamic or highly similar gestures due to the static nature of input images. This limitation could be addressed by integrating temporal features (e.g., via recurrent or 3D convolutional networks) or using attention mechanisms to reinforce focus on discriminative regions. In summary, the use of explainability tools validated the model's correct focus on gesture areas and provided actionable insights for improving ASL recognition. Future work will explore hybrid and temporal modelling approaches guided by these findings, aiming for more robust, generalizable, and interpretable gesture recognition systems.

In addition, this work highlights the value of combining Grad-CAM, LIME, and Integrated Gradients to provide complementary insights into model behavior. These explainability tools not only validated that the models focused on semantically meaningful regions but also revealed their limitations in challenging cases, such as distinguishing between 'm' and 'n'. This multi-technique strategy supports more reliable model auditing and contributes to a deeper understanding of both successes and failures.

## Acknowledgments

We gratefully acknowledge the support of **Begüm Aksoy** and **Piotr Kluczyński** for their assistance during the explainability phase of this study.

## Author contributions

**Methodology:** Fatima-Zahrae EL-Qoraychy, Yazan Mualla.

**Supervision:** Yazan Mualla, mahjoub dridi, Jean-Charles Créput.

**Visualization:** Fatima-Zahrae EL-Qoraychy.

**Writing – original draft:** Fatima-Zahrae EL-Qoraychy.

**Writing – review & editing:** Fatima-Zahrae EL-Qoraychy, Yazan Mualla, Hui Zhao, mahjoub dridi, Jean-Charles Créput, Luca Longo.

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
