## [Decision Letter · Decision Letter 0]

21 Aug 2025

PONE-D-25-36648Explainable AI for sign language recognition models: integrating Grad-Cam, LIME and Integrated GradientsPLOS ONE

Dear Dr. Longo,

Thank you for submitting your manuscript to PLOS ONE. After careful consideration, we feel that it has merit but does not fully meet PLOS ONE’s publication criteria as it currently stands. Therefore, we invite you to submit a revised version of the manuscript that addresses the points raised during the review process.

We look forward to receiving your revised manuscript.

Kind regards,

Marco Antonio Moreno-Armendariz, Ph.D.

Academic Editor

PLOS ONE

Journal Requirements:

Additional Editor Comments:

Please address all the reviewers' comments.

Reviewers' comments:

Reviewer's Responses to Questions

**Comments to the Author**

1. Is the manuscript technically sound, and do the data support the conclusions?

Reviewer #1: Yes

Reviewer #2: Yes

2. Has the statistical analysis been performed appropriately and rigorously?

Reviewer #1: Yes

Reviewer #2: Yes

3. Have the authors made all data underlying the findings in their manuscript fully available?

Reviewer #1: Yes

Reviewer #2: Yes

4. Is the manuscript presented in an intelligible fashion and written in standard English?

Reviewer #1: Yes

Reviewer #2: Yes

5. Review Comments to the Author

Reviewer #1: You refer to a real-time system, so it is necessary to add metrics such as Frames per Second.

When you comment on the model's failure with dynamic signs, how do you manage the different movements of the signs?

At line 56, you mention explainability techniques for Artificial Intelligence models. Please include citations where these mechanisms have been used in other domains.

The manuscript could be improved by including a table detailing the number of images utilized from each dataset.

Another important piece of information missing is the range of random rotation, the percentage of images used for data augmentation, and the type of data augmentation (online or offline).

Reviewer #2: This manuscript presents a timely and valuable study on enhancing the explainability of sign language recognition (SLR) models. Integrating three distinct XAI methods (Grad-CAM, LIME, and Integrated Gradients) to interpret and compare RGB and mask-based CNN models is a significant strength. The work is well-motivated, addresses a clear gap in the literature, and has the potential to contribute meaningfully to the fields of accessible technology and explainable AI. However, several major points require attention to strengthen the methodological rigor, presentation clarity, and analysis depth before it can be considered for publication.

-A significant omission is the absence of a performance comparison with the original VGG19 model from Damion Joyner, which served as the starting point. The paper claims to have enhanced robustness, but without showing the new model's performance on the original "out-of-distribution" tests that the old model failed, this claim is not quantitatively substantiated.

- The real-time prediction interface is mentioned, but its evaluation is superficial. How was it tested? With how many users or in what conditions? Were latency or frame rates measured? Including quantitative results or a user study snippet would greatly strengthen this section.

- What was the final performance (IoU, Dice) of the trained U-Net segmentation model on the HGR1 test set? How exactly was the early stopping criterion defined (e.g., "patience" parameter)? The description of the fusion approach in the original model is criticized, but it's unclear if the new models use any fusion or just two separate streams.

Required Revisions:

1. Add a quantitative comparison with the original baseline model's performance on its failure cases.

2. Provide performance metrics for the U-Net segmentation model.

3. Expand the evaluation of the real-time prediction interface with metrics or a brief user-testing summary.

4. Consider adding a quantitative evaluation of the XAI explanations to supplement the excellent qualitative analysis.

5. Address minor language errors and ensure figure captions are correct.

6. Strengthen the conclusion by explicitly listing the key XAI findings and briefly discussing ethical considerations.

7. Most figures' resolution is not good.

8. How does your work affect Big Data? It is better to add a section related to big data.

8. Include new references between 2024 and 2025. Below are recommended references;

- SignExplainer: an explainable AI-enabled framework for sign language recognition with ensemble learning

- Deep Neural Network-Based Sign Language Recognition: A Comprehensive Approach Using Transfer Learning with Explainability

- FedXAI-ISL: Explainable artificial intelligence-based federated model in recognition and community decentralized learning of Indian sign language

- The AI-powered evolution of big data

- Explainable Real-Time Sign Language to Text Translation

6. PLOS authors have the option to publish the peer review history of their article (what does this mean?). If published, this will include your full peer review and any attached files.

Reviewer #1: No

Reviewer #2: No

---

## [Author Response · Author response to Decision Letter 1]

8 Oct 2025

Manuscript Title:

Explainable AI for sign language recognition models: integrating Grad-Cam, LIME and Integrated Gradients

Manuscript ID PONE-D-25-36648

Authors: Fatima-Zahrae El-Qoraychy, Yazan Mualla, Hui Zhao, Mahjoub Dridi, Jean-Charles Créput, Luca Longo

Dear Editors-in-Chief,

We would like to thank you and the reviewers for the comments on our article. We sincerely appreciate that our article would be of interest to the readers of the PLOS ONE journal. The article has been updated according to the comments of the reviewers.

In the rest of this document, we answer the reviewers’ comments and present a list of the changes (point by point). The reviewers’ comments are in the boxes (in black color). The author’s responses are in colored fonts. A color font is introduced, per reviewer, in the letter and the paper to differentiate the changes between the previous version and the new version of the paper.

Luca Longo,

(on behalf of the authors)

Reviewers' comments:

FormeReviewer #1:

Dear Reviewer #1,

We would like to thank you for your valuable comments and remarks. We answer each of them in the following letter. The changes related to your comments are applied to the new version of the paper with a green colored text. Moreover, in case your comments fit with a comment of another reviewer, the updated text is colorized in purple in the new version of the paper.

R1-1

You refer to a real-time system, so it is necessary to add metrics such as Frames per Second.

We value this suggestion. We have now evaluated the real-time interface. On our hardware, the system achieved an average of 10 FPS with a mean latency of 100 ms per frame (see Section Real-Time Evaluation).

R1-2

When you comment on the model's failure with dynamic signs, how do you manage the different movements of the signs?

We thank the reviewer for raising this point. While 'j' and 'z’ are dynamic ASL signs, some of the datasets we employed provide static image approximations of these gestures. We included these approximations to allow coverage of the full 26-letter ASL alphabet. We clarify this in the Methods section 3.3.1, and in the Discussion section we acknowledge the limitation that such static representations cannot fully capture the temporal nature of these signs. We also indicate that future work will extend our framework to spatio-temporal models better suited for dynamic gestures.

R1-3

At line 56, you mention explainability techniques for Artificial Intelligence models. Please include citations where these mechanisms have been used in other domains.

Thanks for this helpful suggestion. In the revised manuscript, we have enriched the introduction with citations showing the successful application of explainability techniques in various domains.

R1-4

The manuscript could be improved by including a table detailing the number of images utilized from each dataset.

We thank the reviewer for this suggestion. Our dataset was constructed by merging several publicly available ASL alphabet datasets. To ensure class balance, all images were mixed and reorganized into a unified dataset of approximately 325,000 samples (12,500 per class). During this preprocessing, the original per-source counts were not preserved. Instead, we now provide the link to the processed dataset in Section 3.3.1 of the revised manuscript.

R1-5

Another important piece of information missing is the range of random rotation, the percentage of images used for data augmentation, and the type of data augmentation (online or offline).

We thank the reviewer for pointing out this important detail. In the revised manuscript, we clarify that no geometric or photometric data augmentation (rotation, shift, zoom, brightness, etc.) was applied. Instead, we only used ImageDataGenerator for preprocessing, with pixel-value rescaling (1/255) and batch generation. This ensured standardized inputs but did not artificially increase data diversity. We also note this limitation in the Discussion and mention that future work may integrate stronger augmentation pipelines to further improve robustness.

We appreciate your valuable comments.

Reviewer #2:

Dear Reviewer #2,

We would like to thank you for your valuable comments and remarks. We answer each of them in the following letter. The changes related to your comments are applied to the new version of the paper with a blue colored text. Moreover, in case your comments fit with a comment of another reviewer, the updated text is colorized in purple in the new version of the paper.

R2-1

This manuscript presents a timely and valuable study on enhancing the explainability of sign language recognition (SLR) models. Integrating three distinct XAI methods (Grad-CAM, LIME, and Integrated Gradients) to interpret and compare RGB and mask-based CNN models is a significant strength. The work is well-motivated, addresses a clear gap in the literature, and has the potential to contribute meaningfully to the fields of accessible technology and explainable AI. However, several major points require attention to strengthen the methodological rigor, presentation clarity, and analysis depth before it can be considered for publication.

-A significant omission is the absence of a performance comparison with the original VGG19 model from Damion Joyner, which served as the starting point. The paper claims to have enhanced robustness, but without showing the new model's performance on the original "out-of-distribution" tests that the old model failed, this claim is not quantitatively substantiated.

- The real-time prediction interface is mentioned, but its evaluation is superficial. How was it tested? With how many users or in what conditions? Were latency or frame rates measured? Including quantitative results or a user study snippet would greatly strengthen this section.

- What was the final performance (IoU, Dice) of the trained U-Net segmentation model on the HGR1 test set? How exactly was the early stopping criterion defined (e.g., "patience" parameter)? The description of the fusion approach in the original model is criticized, but it's unclear if the new models use any fusion or just two separate streams.

Required Revisions:

1. Add a quantitative comparison with the original baseline model's performance on its failure cases.

We thank the reviewer for this valuable comment. We initially considered including a quantitative comparison with the original baseline model (VGG19, Joyner), but the baseline failed to provide usable predictions on the datasets we employed. This limitation was already highlighted by the original author on the project website, where it is reported that the model struggles significantly with out-of-distribution samples. As a result, a direct numerical comparison was not meaningful. Instead, in our revised manuscript, we clarify this limitation and emphasize how our proposed models overcome this issue by producing consistent and interpretable results on the same test conditions.

R2-2

2. Provide performance metrics for the U-Net segmentation model.

Thanks for this valuable suggestion. In the revised manuscript, we have expanded Section 3.3.2 to provide a detailed description of the U-Net segmentation model, including its architecture, training configuration, and role within our framework.

R2-3

3. Expand the evaluation of the real-time prediction interface with metrics or a brief user-testing summary.

We value this suggestion. We have now evaluated the real-time interface. On our hardware, the system achieved an average of 10 FPS with a mean latency of 100 ms per frame (see Section Real-Time Evaluation).

R2-4

4. Consider adding a quantitative evaluation of the XAI explanations to supplement the excellent qualitative analysis.

We appreciate this valuable suggestion. In the revised version, we have added a quantitative analysis of the saliency maps produced by Grad-CAM, LIME, and Integrated Gradients. Two complementary metrics were computed: (i) energy concentration, defined as the proportion of attribution energy captured by the top-k most salient pixels (k=5,10,20), and (ii) entropy, which measures the overall dispersion of the saliency map. These metrics were first calculated for six representative ASL letters (a, l, i, j, m, n) on both RGB- and mask-based models, and then averaged to provide a global comparison.

R2-5

5. Address minor language errors and ensure figure captions are correct.

We carefully proofread the manuscript and corrected typographical issues. We also standardized figure captions.

R2-6

6. Strengthen the conclusion by explicitly listing the key XAI findings and briefly discussing ethical considerations.

We appreciate this constructive suggestion. In the revised manuscript, we have strengthened the Conclusion by explicitly listing the key findings related to the integration of XAI. In particular, we highlight (i) how the three complementary techniques (Grad-CAM, LIME, and Integrated Gradients) provide spatial, local, and structural perspectives, (ii) how explainability helped us identify both correct model focus and failure cases (e.g., distinguishing m vs. n), and (iii) how this multi-technique strategy supports more reliable model auditing. In addition, we have added a brief note on ethical considerations, emphasizing that transparency and trust are critical in assistive technologies, particularly for accessibility and fairness in communication between hearing and deaf communities.

R2-7

7. Most figures' resolution is not good.

We thank the reviewer for pointing this out. In the revised version of the manuscript, the figures have been replaced with higher-resolution versions to improve clarity and readability.

R2-8

8. How does your work affect Big Data? It is better to add a section related to big data.

We thank the reviewer for this valuable suggestion. In the revised manuscript, we have added a subsection entitled “Implications for Big Data”. This section briefly outlines how our framework can contribute to transparency and interpretability when dealing with large-scale and heterogeneous datasets, which are increasingly common in Big Data scenarios.

R2-9

9. Include new references between 2024 and 2025. Below are recommended references;

- SignExplainer: an explainable AI-enabled framework for sign language recognition with ensemble learning

- Deep Neural Network-Based Sign Language Recognition: A Comprehensive Approach Using Transfer Learning with Explainability

- FedXAI-ISL: Explainable artificial intelligence-based federated model in recognition and community decentralized learning of Indian sign language

- The AI-powered evolution of big data

- Explainable Real-Time Sign Language to Text Translation

We appreciate this helpful suggestion. In the revised manuscript, we have ensured the inclusion of recent and relevant works from 2024–2025. Specifically, we note that two of the recommended references (SignExplainer and Deep Neural Network-Based Sign Language Recognition: A Comprehensive Approach Using Transfer Learning with Explainability) were already cited in the previous version of the manuscript. We have now added the remaining suggested references, namely FedXAI-ISL: Explainable artificial intelligence-based federated model in recognition and community decentralized learning of Indian sign language, The AI-powered evolution of big data, and Explainable Real-Time Sign Language to Text Translation. These additions strengthen the manuscript by broadening its coverage of recent advances in explainable sign language recognition and data-intensive AI applications.

We appreciate your valuable comments.

---

## [Decision Letter · Decision Letter 1]

28 Oct 2025

Explainable AI for sign language recognition models: integrating Grad-Cam, LIME and Integrated Gradients

PONE-D-25-36648R1

Dear Dr. Longo,

We’re pleased to inform you that your manuscript has been judged scientifically suitable for publication and will be formally accepted for publication once it meets all outstanding technical requirements.

Kind regards,

Marco Antonio Moreno-Armendariz, Ph.D.

Academic Editor

PLOS ONE

Additional Editor Comments (optional):

Dear Author,

I am satisfied with the quality of the manuscript. Please fill in the comments of the reviewers in your final version of your manuscript.

Reviewers' comments:

Reviewer's Responses to Questions

**Comments to the Author**

1. If the authors have adequately addressed your comments raised in a previous round of review and you feel that this manuscript is now acceptable for publication, you may indicate that here to bypass the “Comments to the Author” section, enter your conflict of interest statement in the “Confidential to Editor” section, and submit your "Accept" recommendation.

Reviewer #1: All comments have been addressed

Reviewer #3: (No Response)

Reviewer #4: (No Response)

2. Is the manuscript technically sound, and do the data support the conclusions?

Reviewer #1: Yes

Reviewer #3: Yes

Reviewer #4: Partly

3. Has the statistical analysis been performed appropriately and rigorously?

Reviewer #1: Yes

Reviewer #3: Yes

Reviewer #4: No

4. Have the authors made all data underlying the findings in their manuscript fully available?

Reviewer #1: Yes

Reviewer #3: Yes

Reviewer #4: Yes

5. Is the manuscript presented in an intelligible fashion and written in standard English?

Reviewer #1: Yes

Reviewer #3: Yes

Reviewer #4: Yes

6. Review Comments to the Author

Reviewer #1: I have no further comments. Thank you for responding to the previous review. This article is ready for publication.

Reviewer #3: As core classification model is explicitly a modified VGG19 architecture, a well-established design. While the application to the new, balanced dataset and the comparison of RGB vs. mask input are valuable, the architectural novelty is minimal.

1. why VGG19 architecture is chosen is currently weakly justified, please clearly articulate why this specific architecture was chosen over more recent, state-of-the-art (e.g., ResNet-50, MobileNetV2) and recent SLR works - is there any architectural advantage for this experiment

2. author also mentioned - real-time performance evaluation (10 FPS with 100 ms latency), please specify the exact model of the Intel CPU used for the measurement

Reviewer #4: 1. The paper gives a good approach using Grad-CAM, LIME, and Integrated Gradients for sign language recognition.

2. More details about the dataset and preprocessing steps are needed.

3. The fusion of RGB and depth data can be improved using advanced methods.

4. Add some quantitative results to support the visual explanations.

5. Improve the quality and clarity of figures.

6. Include a short discussion on ethical and fairness aspects.

7. Explain how dynamic signs with motion can be handled in future work.

8. Strengthen the conclusion with main results and future applications.

9. Check grammar, formatting, and figure captions carefully.

10. Add a few recent papers (2024–2025) to update the literature review.

7. PLOS authors have the option to publish the peer review history of their article (what does this mean?). If published, this will include your full peer review and any attached files.

Reviewer #1: **Yes: **Jaime Arturo Lara Cázares

Reviewer #3: No

Reviewer #4: No

---

## [Editor Report · Acceptance letter]

PONE-D-25-36648R1

PLOS ONE

Dear Dr. Longo,

I'm pleased to inform you that your manuscript has been deemed suitable for publication in PLOS ONE. Congratulations! Your manuscript is now being handed over to our production team.

Kind regards,

on behalf of

Professor Marco Antonio Moreno-Armendariz

Academic Editor

PLOS ONE